# Effects of Water Column Variations on Sound Propagation and Underwater Acoustic Communications

**DOI:** 10.3390/s19092105

**Published:** 2019-05-07

**Authors:** Salman I. Siddiqui, Hefeng Dong

**Affiliations:** Department of Electronic Systems, Norwegian University of Science and Technology (NTNU), NO-7491 Trondheim, Norway; hefeng.dong@ntnu.no

**Keywords:** underwater sound propagation, underwater channel variability, underwater acoustic communications

## Abstract

Underwater sound propagation is very sensitive to geometric and environmental variations. The geometric variations are the motion of the source and/or receiver, while the environmental variations are due to surface motion and water column variations. In order to reduce the effects of these variations, it is necessary to understand their effects on sound propagation. In this paper, some water column variations are reported, and their effect on the underwater sound propagation is studied. These water column variations were observed during an experiment in the TrondheimFjord on 22 September 2016. Strong amplitude variations were observed in the channel impulse response during the experiment. The Doppler analysis was performed on the channel impulse response, which showed strong Doppler variations. The amplitude and Doppler variations suggested the presence of water column variations. To demonstrate the performance of the communication system, the time reversal combiner was implemented. The system performance was demonstrated by computing the mean squared error between the transmitted signal and the output of the combiner. The performance of the combiner degraded by ~2.5 dB in the presence of water column variations. Due to these variations, the amplitudes of the arrivals changed rapidly, which posed a great challenge for the time reversal combiner. These amplitude variations affected the focusing of the combiner and hence induced intersymbol interference and performance degradation. This work provides an insight into the effects of water column variations on underwater sound propagation and underwater acoustic communications.

## 1. Introduction

The underwater acoustic channel is a bounded medium with the sea surface and the sea bottom. In this bounded medium, the acoustic signals propagate through multiple paths including the direct path, surface reflected path, bottom reflected path, and other paths with multiple interactions from the surface and/or bottom. These different paths result in time spreading of the acoustic signal, which causes multipath interference. The acoustic signal is also affected by the Doppler induced by the motion of the source and/or receiver, surface variations, and water column variations. The multipath and Doppler affect the acoustic signal propagation and acoustic communications. This paper reports some water column effects that were observed during an experiment and studies their effects on the underwater sound propagation and underwater acoustic communications.

Underwater sound propagation is very sensitive to the water column properties like the sound speed profile (SSP). The SSP is a function of water temperature, salinity, and depth. It also changes with time and space, which causes time variability in sound propagation. These time variabilities can be due to many different physical phenomena happening in the water column such as internal waves, eddies, internal tides, or solitons. Internal waves are usually studied from synthetic aperture radar (SAR) images [1]. Different image processing techniques have been applied on SAR images to study internal waves in a more effective way [2]. Internal waves were observed in Norwegian waters and reported in [3]. In [3], these waves were characterized by a relative difference in the pixel value between crest and trough, wave width, and the number of crests. The typical pixel difference of 5 dB, wave width of 11 km, and four wave crests have been reported for Norwegian waters.

The effects of these water column variations on the underwater sound propagation have been studied in the literature. In [4], an anomalous behavior was observed in the frequency response, which was caused by the interaction with the internal waves. In [5], the SSP variations were reported in the middle of the water column, which was caused by the internal waves. In [6], an internal wave bolus was detected in an SAR image. It was shown that internal waves introduced temperature variations in the water column, which resulted in the bending of the acoustic rays. The effects of water column variations on geo-acoustic inversion have also been studied [7,8,9]. In [7], shallow water sound propagation was modeled for a few kilometers in the presence of water column variations, and it was reported that the water column variations severely affected the estimated seabed properties. In [8], a transform method was presented to model the effects of water column variations on the acoustic inversion. In [9], the effects of internal waves on the geo-acoustic inversion were presented. It was shown that internal waves induced focusing and de-focusing of the sound intensity, which greatly affected the inversion results.

In this paper, the effects of water column variations on sound propagation and underwater acoustic communications are studied. In the literature, the physical phenomena in the water column have been studied using SAR images or thermistor chains. In this work, channel impulse response (IR) analysis is performed to show water column variations. In [10], the effect of water column uncertainties on the non-coherent communication system was studied. The results showed that the system performance improved in the presence of the thermocline. The work presented in this paper is different from [10] in a number of ways. Firstly, the time scale of these variations is different. The water column variations, observed in [10], were on the scale of a few hours, while in this paper, the variations were observed on the scale of a few minutes. Secondly, the source-receiver positions and ranges were very different. In [10], the source was deployed at the bottom, and the source-receiver range was 3 km. In this paper, the source and receiver were deployed in the middle of the water column, and the source-receiver ranges were 480 m and 670 m. Due to source-receiver positioning, different effects were observed and discussed.

In order to study the effects of water column variations on sound propagation, the channel IR estimates were studied. Two distinct features were observed in the arrival pattern: (1) the relative amplitudes of the first and second arrival were different at different receivers, and (2) strong amplitude variations were observed at the receivers due to the interaction between the acoustic signal and the water column variations. The Doppler analysis was also performed on the channel IR, which provided useful insight into the propagation of the water column variations. It was observed that a strong Doppler was induced due to the interaction between the receiver and the water column variations. The channel variability due to water column variations was presented by channel coherence analysis. The channel coherence along time at each receiver reduced due to time variability in the channel IR. The effect of water column variations on underwater communication was also demonstrated by a time reversal (TR) combiner. In the TR system, a known signal was transmitted for channel IR estimation. Assuming that all the multipath were cleared, the data signal was transmitted. The received data signal was correlated with the time-reversed version of the estimated channel IR. The performance of the TR combiner degraded in the presence of water column variations, which shows that these variations affect underwater communication and that there is a need for a robust technique to combat these variations. The main work of this paper is: (1) to study the effects of water column variations on underwater sound propagation using channel IR analysis, (2) to study the effects of these variations on underwater acoustic communications.

The paper is organized as follows: The description of the experiment is presented in Section 2. Section 3 describes the experimental data and results. It shows the channel IR analysis, channel coherence analysis, and the performance of the underwater communication system. Section 4 concludes the paper.

## 2. Experiment Description

This section presents the description of the experiment, which was performed on 22 September 2016 in TrondheimFjord. During the experiment, four autonomous hydrophones were used. Figure 1 shows the location of the source (Position B) and the receivers (Positions C and D) and the water depths at these positions. The source was deployed at a 20-m depth, and the water depth at the source position was 150 m. Two autonomous hydrophones were deployed at both Positions C and D in an array formation at 30 m and 60 m depth. The water depth was 200 m at C and 100 m at D. The source-receiver ranges at Positions C and D were 480 m and 670 m, respectively. Two sound speed profiles (SSPs) were obtained at different times at Positions B, C, and D and shown in Figure 2.

The source signal was transmitted from the transmitter deployed on the NTNU research vessel R/V Gunnerus. Different sets of acoustic signals were transmitted during the experiment. The channel IR was estimated from the chirp signals, and the data signal was transmitted in the form of a binary phase shift keying (BPSK) modulated signal. In this paper, the signals between 14:11 and 14:20 were analyzed. The chirp signals were transmitted in the frequency band of 6320–10,680 Hz, and the data signal was transmitted in the 7000–10,000 Hz band. A total of 50 chirps were transmitted each minute where each chirp was 0.1 s long, and a silence of 0.2 s was placed between each chirp to clear the multipath. The data signal was transmitted for 40 s after the chirp signal transmission, and 5 s of silence was placed at the end of each minute. The same sequence was transmitted for 9 min.

## 3. Experimental Data Analysis and Results

In this section, the effects of water column variations on underwater signal propagation and communication are discussed. This section is divided into three parts. In the first part, the channel IR estimates are studied and some of the environmental effects on the channel IR estimates are explained. In the second part, the time variability is studied in the channel IR estimates by computing channel coherence. In the third part, the performance of the underwater communication system is shown at both Positions C and D.

### 3.1. Channel IR Analysis

In this section, the channel IR estimates from the experimental data are studied at both receiver positions. Figure 3 shows the amplitude of the first two arrivals of channel IR estimates at Position D between 14:11 and 14:20 at depths of 30 m (top panel) and 60 m (bottom panel), respectively. There were gaps of 45 s in each minute because the channel IR estimates were obtained from the chirp signals, which were only transmitted during the first 15 s in each minute. Two observations are made on this figure. Firstly, the relative amplitude of the two arrivals was very different at both receivers. The first arrival had a higher amplitude at 30 m depth, while the second arrival had a higher amplitude at 60 m depth. Secondly, there were strong amplitude variations in the second arrival between 14:13 and 14:14 at 30 m depth and between 14:14 and 14:15 at 60 m depth.

The Doppler analysis was performed on the channel IR estimates using the time windowed Doppler spectrum (TWDS) technique [11]. Using TWDS, it is possible to study the temporal evolution of Doppler, which helps in understanding the effects of environmental variations on the acoustic signals. In this technique, the Doppler analysis was done on the time window of channel IR, and this time window was slid over the complete channel IR. In this analysis, the time window of 1.5 s was selected. Figure 4 shows the Doppler induced in the first two arrivals at both receivers at Position D in three intervals 14:11–14:12, 14:14–14:15, and 14:17–14:18, respectively. The top panel shows the Doppler variations in the first arrival, and the bottom panel shows the Doppler variations in the second arrival. The left and right panels show the Doppler variations at 30 m and 60 m depths, respectively. The positive and negative values in the Doppler were the contraction and expansion in the path between the source and the receiver. Firstly, the Doppler variations were similar at the first and second arrival in all the figures, which shows that the surface variations did not induce any Doppler. The wind speed was around 1 m/s, and the sea surface was very calm during the day of the experiment. Secondly, a different Doppler was induced at different instants. The Doppler variations were much higher at both 30 m and 60 m depths between 14:14 and 14:15. Thirdly, the Doppler variations were much higher at 60 m depth than at 30 m depth, which suggests that the deeper receiver was more affected by the water column variations.

Figure 5 shows the Doppler variations during 9 min of acoustic signal transmission at Position D at 30 m depth (left panel) and 60 m depth (right panel). The top panel shows the Doppler variations in the first arrival, and the bottom panel shows the Doppler variations in the second arrival. Each bar represents the Doppler variation during 15 s of chirp signal transmission, e.g., the bar at 14:11 shows the Doppler variations between 14:11 and 14:12. A strong Doppler was induced between 14:14 and 14:15 in both arrivals at 30 m and 60 m depths. The strong Doppler and amplitude variations occurred at the same time at 60 m depth, but not at 30 m depth (Figure 3). At 30 m depth, the amplitude variations occurred one minute before the Doppler variations. It was expected that the amplitude variations were due to the interaction between the acoustic signal and the water column variations. The water column variations interacted with the receivers at Position D at 14:15, which induced strong Doppler at both receivers.

Figure 6 shows the amplitude of the first two arrivals at Position C at 30 m depth (top panel) and at 60 m depth (bottom panel). At 30 m depth, the second arrival had a higher amplitude at the start, but the amplitude of the first arrival increased over time and crossed the second arrival at 14:16. At 60 m depth, the amplitudes of the two arrivals were very close to each other. There were strong amplitude variations between 14:16 and 14:20 at 30 m depth and between 14:15 and 14:20 at 60 m depth.

Figure 7 shows the Doppler variations at both arrivals and both receivers at Position C. The left and right panels show the Doppler variations in the 30 m and 60 m depth receivers, respectively. The top panel shows the Doppler variations in the first arrival, and the bottom panel shows the Doppler variations in the second arrival. There were no strong Doppler variations observed in any of the arrivals at Position C, which suggests that the water column variations did not interact with the receiver at Position C. On the other hand, there were strong amplitude variations at both receivers at Position C (Figure 6). This suggests that the water column variations traveled between the source-receiver transect B-C and that the amplitude variations were induced due to the interaction between the acoustic signal and the water column variations.

Based on the amplitude and Doppler variations, it is possible to suggest the propagation of the water column variations. It is impossible to find the exact propagation angle of the water column variations because they only interacted with Position D. Figure 8 shows the movement of the water column variations along the range. The white line represents the water column variations traveling in the water, and the white arrows show its traveling direction. The water column variations entered the transect B-D between 14:13 and 14:14 and interacted with both receivers at Position D between 14:14 and 14:15. No strong Doppler or amplitude variations were observed at Position D after 14:15. At Position C, the strong amplitude variations started at 60 m depth between 14:15 and 14:16 and at 30 m depth between 14:16 and 14:17. The amplitude variations continued till 14:20 at both receivers. The amplitude variations suggest that the acoustic signal interacted with the water column variations from 14:15 till 14:20 as it traveled towards Position C. No strong Doppler variations were observed at the receivers at Position C; therefore, the water column variations did not reach Position C till 14:20.

### 3.2. Channel Coherence Analysis

The channel coherence provides a link between the water column variations and its effect on the communication system. Strong amplitude variations were observed in the channel IR estimates due to the interaction between the water column variations and the acoustic signal.

This section shows the channel coherence between the estimated channel IRs from the chirp signals. The channel coherence expresses the channel variability over time. It was computed as the product of the first channel IR (considered as reference channel IR) with the channel IRs computed from all chirp signals transmitted during 9 min of transmission. The main-lobe represents the correlation between the highest amplitude arrivals, and the side-lobes represent the contribution of the multipath interference. The multipath interference was expressed as the ratio of the energy of the side-lobes and the energy of the main-lobe.

Figure 9 shows the temporal channel coherence from 14:11–14:20 at Position D at 30 m depth (left panel) and at 60 m depth (right panel). For both figures, the reference channel IR was considered at 14:11, and the coherence is shown with the channel IR computed from the first chirp signal every minute till 14:20. There were small amplitude variations in the side-lobes at 30 m depth. At 60 m depth, the channel variations were stronger, which can be seen in the channel coherence. The amplitude of the side-lobes was higher at 14:14 and 14:18, which can be explained by the amplitude of the arrivals in Figure 3. At 30 m depth, the amplitude of the first arrival was higher than the second arrival during the 9 min of the received signal. Therefore, the main-lobe was the cross-correlation of the first arrival. At 60 m depth, the amplitude of the first arrival exceeded the second arrival at 14:14 and 14:18. When the highest amplitude arrival switched from the first arrival to the second arrival, the main-lobe was obtained by the contribution of two different arrivals, which resulted in incoherent summation of the paths. Figure 10 represents the temporal coherence of the TR combiner at Position D (left panel). At 14:11, the amplitude of the main-lobe was enhanced, and the side-lobes were suppressed; however, at 14:13, the amplitude of the side-lobes was enhanced, and main-lobe was suppressed. A similar effect appeared at 14:19 also. Due to an increase in the amplitude of the side-lobes, the interference increased, which degraded the performance of the communication system. Figure 10 also shows the multipath interference at both receivers and the combiner for all the 15 s of chirp transmission each minute (right panel). The multipath interference at each receiver increased over time due to channel variations. The TR reduced the multipath interference at the start and between 14:18 and 14:19, but the interference increased between 14:13 and 14:15 and between 14:19 and 14:20. The high interference between 14:13 and 14:15 was because of the water column variations. There were no amplitude variations observed between 14:19 and 14:20; therefore, the high interference at this instant needs to be further investigated.

A similar analysis was done for Position C. Figure 11 represents the temporal coherence of the TR combiner (left panel). The amplitude of the side-lobes was high at 14:12, 14:15, and 14:18, which increased the interference. Figure 11 also shows the multipath interference at each receiver and the combiner for the chirp transmission of 15 s during each minute (right panel). The multipath interference increased slowly at each receiver over time, which was due to channel variations. The multipath interference was very high for combiner output at 14:12, 14:15, and 14:18 due to non-coherent addition, which resulted in the suppression of the main-lobe and enhancement of the side-lobes. The high multipath interference at 14:15 and 14:18 was due to strong amplitude variations and water column variations, but the amplitude variations observed at 14:12 were not very high. The trend of multipath interference was periodic with a period of 3 min, which suggests that the high multipath interference might be due to water column variations, but it needs to be further investigated.

### 3.3. Communication Data Analysis

This section shows the effect of water column variations on the communication system. A TR communication system was implemented in this paper. Figure 12 shows the block diagram of the TR system. In the block diagram, δ(t) is the pulse signal, which was transmitted for channel IR estimation $g_{i}\left(t=0\right)$. The data signal x(t) was first pulse shaped and then convolved with the channel IRs $g_{i}\left(t\right)$, which was the channel during the signal transmission. The noise signals u(t) and w(t) were added to the channel IR estimate and data signal and then sent to the TR combiner block. The TR combiner block performed the TR operation by convolving the channel IR estimate with the data signal to get the output $z_{output}$. The MSE was computed between the input data signal x(t) and $z_{output}$ and used as a performance metric to study the effects of water column variations on the communication system.

Figure 13 shows the degradation of the performance of the communication system at Position D (left panel) and Position C (right panel), respectively. The data were divided into blocks of 0.25 s, so each data point in the figure represents the MSE for 0.25 s of data transmission. To study the channel variations for the whole 9 min, the reference channel IR estimate was considered at 14:11. At Position D, the performance of the system was very good at the start, which indicates a good match between the channel IRs. After the first minute, the performance degraded due to strong amplitude variations. The performance degraded again between 14:18 and 14:19. At Position C, the performance variations were much faster as compared to that at Position D. The MSE performance degraded at 14:12, 14:15, and 14:18, which is consistent with the multipath interference shown Figure 10.

Figure 14 shows the mean MSE at Positions C and D (left panel). The mean value was calculated each minute by averaging the MSE values of one minute. During the second and third minute at Position D, the performance of the TR system degraded by 2 dB and 2.5 dB, respectively. At 14:18, the performance degraded by 1.7 dB. At Position C, the mean MSE degraded by 3 dB every 3 min. Figure 14 shows the mean BER for each minute of transmission (right panel). At both Positions C and D, the mean BER increased by almost 20–40% in the presence of water column variations. These results show that the water column variations have a strong impact on the performance of the communication system.

## 4. Conclusions

In this paper, the effect of water column variations on the sound propagation and acoustic communications in TrondheimFjord was studied. The experiment was performed on 22 September 2016. During the experiment, strong amplitude variations were observed in the channel IR estimates due to the interaction between the water column variations and the acoustic signal. In addition to the amplitude variations, strong Doppler variations were induced due to the interaction of the water column variations with the receivers. These variations affected the channel coherence along time and the performance of the communication system. To demonstrate the effect of water column variations on communications, a TR system was implemented. It was shown that the MSE of the TR communication system degraded by 2.5 dB and that the BER increased by 30% due to water column variations. This study shows that water column variations have a significant effect on sound propagation and underwater acoustic communications, which requires an improved communication system. 

## Figures and Tables

**Figure 1 sensors-19-02105-f001:**
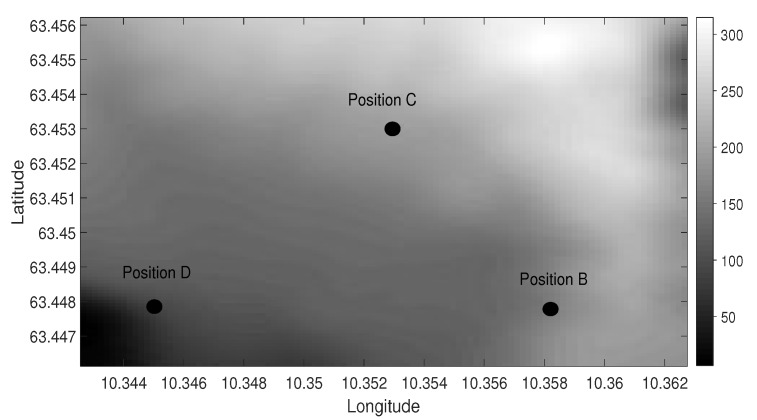
Bathymetry diagram of the experiment site. Position B shows the source position, and Positions C and D show the receiver positions. The water depth at the source position was 150 m. The water depths at the receiver positions were 100 m and 200 m. The source-receiver ranges at Positions C and D were 480 m and 670 m, respectively.

**Figure 2 sensors-19-02105-f002:**
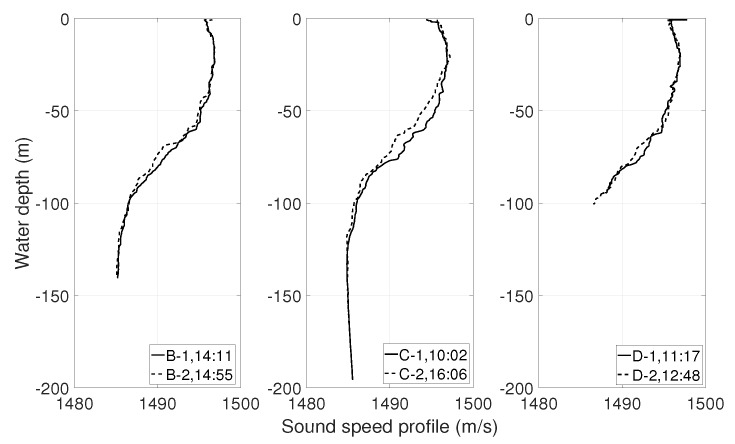
Sound speed profiles (SSPs) obtained at Positions B (left panel), C (middle panel), and D (right panel). At Positions B, C, and D, SSPs were collected at 14:11 and 14:55, 10:02 and 16:06, and 11:17 and 12:48, respectively.

**Figure 3 sensors-19-02105-f003:**
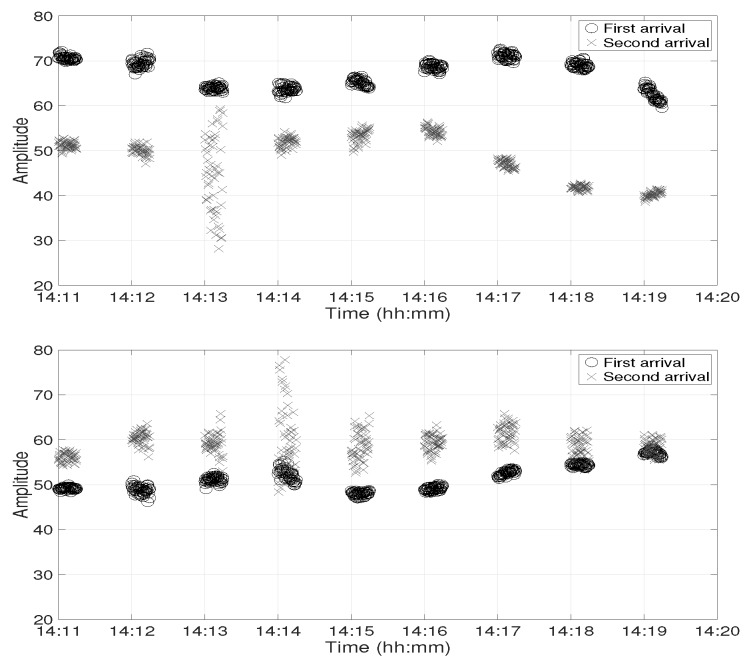
Amplitude of the first two arrivals in the channel impulse response (IR) computed from the experimental data at Position D. The circle and cross markers represent first and second arrivals at 30 m depth (**top panel**) and at 60 m depth (**bottom panel**), respectively.

**Figure 4 sensors-19-02105-f004:**
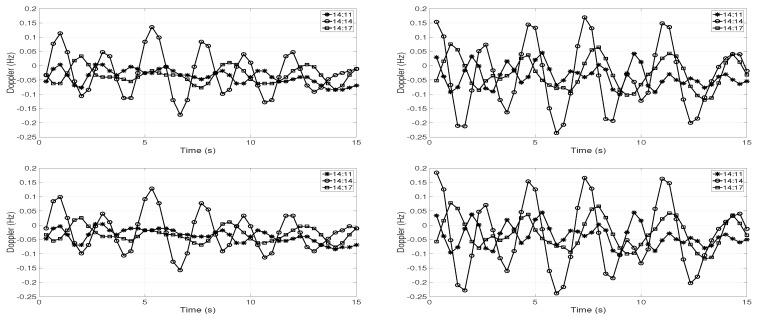
Doppler variations in the channel IR estimates at Position D at three different time instants; 14:11–14:12, 14:14–14:15, and 14:17–14:18. The top panel shows the Doppler variations in the first arrival, and the bottom panel shows the Doppler variations in the second arrival. The left and right panels show the Doppler variations at 30 m and 60 m depths, respectively.

**Figure 5 sensors-19-02105-f005:**
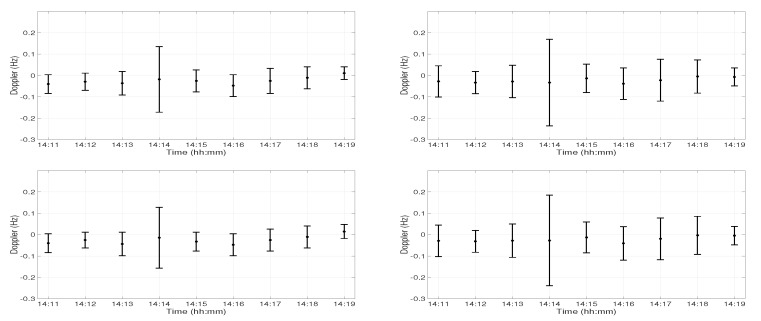
Doppler variations during 9 min of acoustic signal transmission at Position D at 30 m depth (**left panel**) and 60 m depth (**right panel**). The top panel shows the Doppler variations in the first arrival, and the bottom panel shows the Doppler variations in the second arrival.

**Figure 6 sensors-19-02105-f006:**
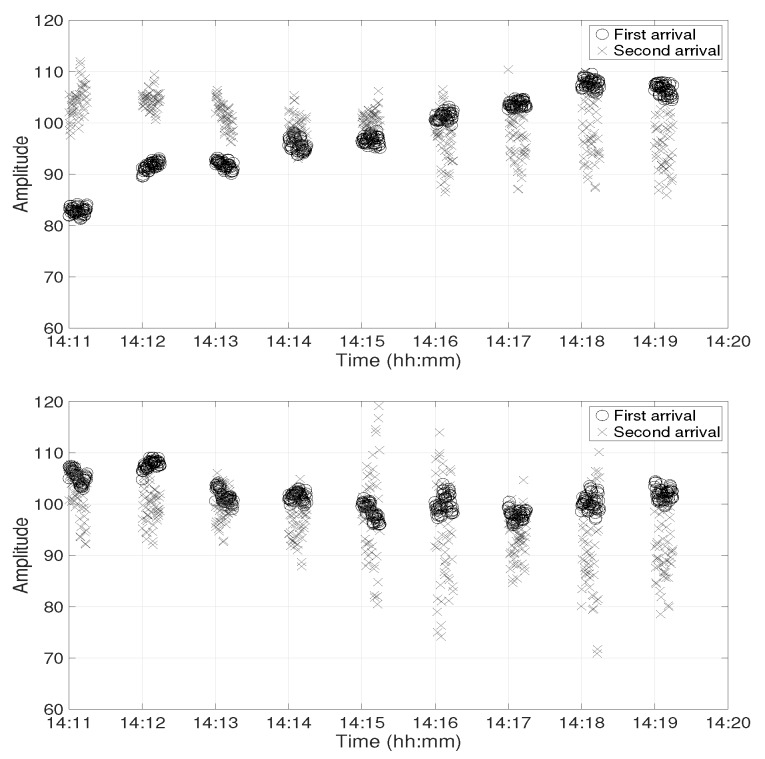
Amplitude of the first two arrivals in the channel IR computed from the experimental data at Position C. The circle and cross markers represent first and second arrivals at 30 m depth (**top panel**) and at 60 m depth (**bottom panel**), respectively.

**Figure 7 sensors-19-02105-f007:**
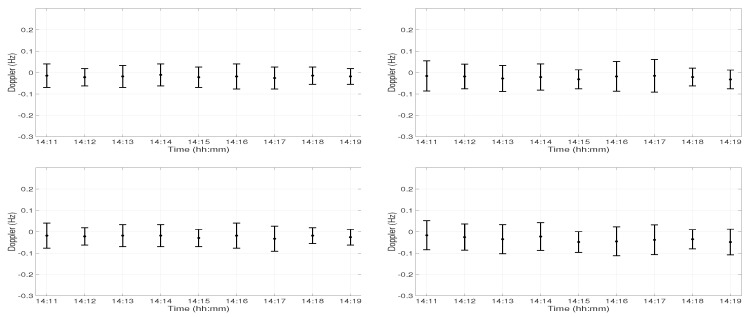
Doppler variations during 9 min of acoustic signal transmission at Position C at 30 m depth (**left panel**) and at 60 m depth (**right panel**). The top panel shows the Doppler variations in the first arrival, and the bottom panel shows the Doppler variations in the second arrival.

**Figure 8 sensors-19-02105-f008:**
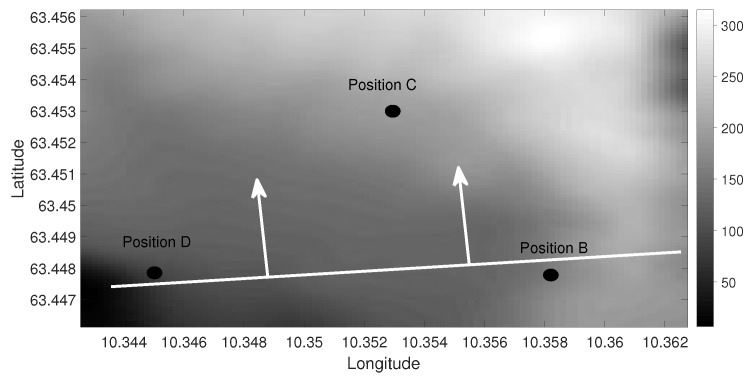
Bathymetry diagram showing the direction of the water column variations along the range. The white line represents the water column variations traveling in the water, and the white arrows show its traveling direction.

**Figure 9 sensors-19-02105-f009:**
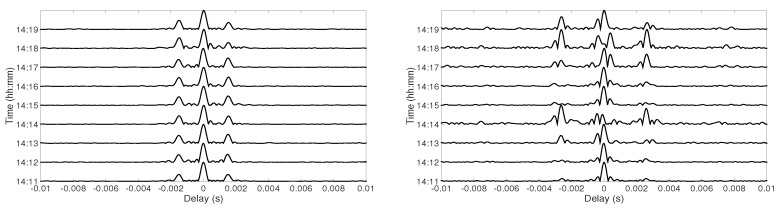
The channel coherence at Position D at 30 m depth (**left panel**) and at 60 m depth (**right panel**), respectively.

**Figure 10 sensors-19-02105-f010:**
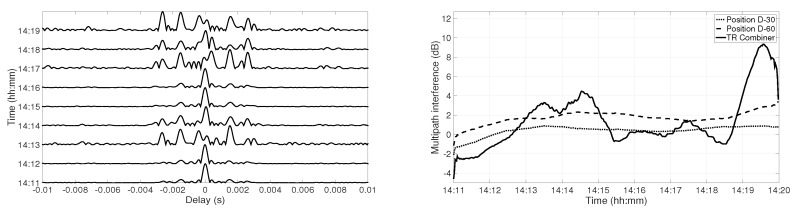
Temporal channel coherence after combining the received signal at 30 m depth and at 60 m depth at Position D (**left panel**); multipath interference at Position D (**right panel**).

**Figure 11 sensors-19-02105-f011:**
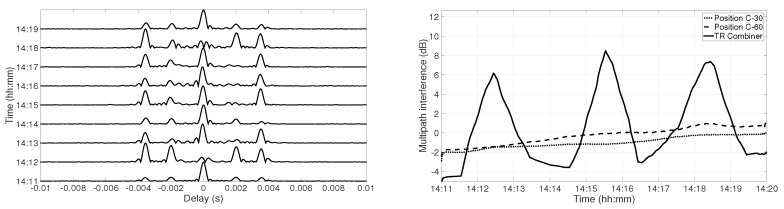
Temporal channel coherence after combining the received signal at 30 m depth and at 60 m depth at Position C (**left panel**); multipath interference at Position C (**right panel**).

**Figure 12 sensors-19-02105-f012:**
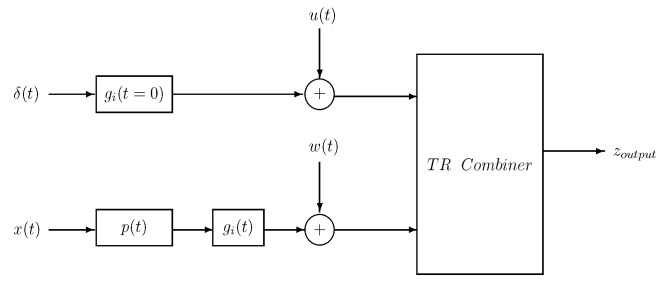
Block diagram of the time reversal (TR) communication system.

**Figure 13 sensors-19-02105-f013:**
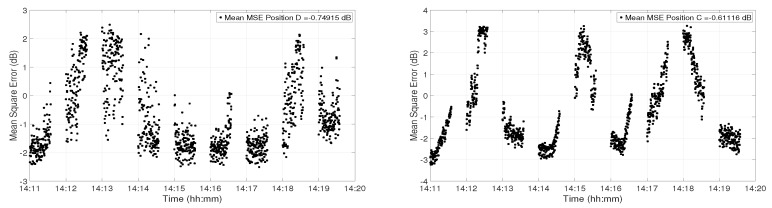
Performance degradation (in terms of MSE) of the communication system due to water column variations for 9 min of the received signal at Position D (**left panel**) and Position C (**right panel**), respectively.

**Figure 14 sensors-19-02105-f014:**
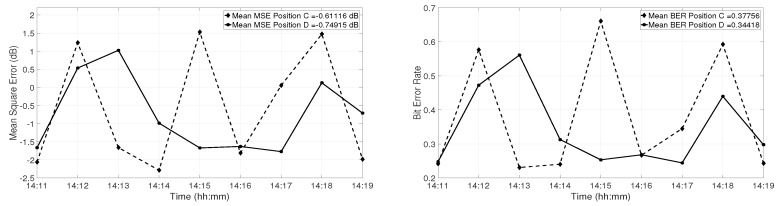
Performance of the TR communication system in terms of MSE averaged over one minute of the received signal (**left panel**) and in terms of BER averaged over one minute of the received signal (**right panel**). The dashed line represents the performance at Position C, and the continuous line represents the performance at Position D, respectively.

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
