# Peer review of "Effects of Water Column Variations on Sound Propagation and Underwater Acoustic Communications"

_sensors, 2019, doi:10.3390/s19092105_

Round 1
Reviewer 1 Report
This is a good paper.
What are the question marks for on line 191? Please fix
One major concern is the water column variations, defined as temperature, salinity and depth.
Is seems that depth was the only factor studied. Is this the case? If not, then data is needed on the temperature and the salinity. If so, then a section saying why they weren't examined or how the depth parameter was isolated is required.
Your conclusion supports the theory that all three parameters effect the communications but without knowing what effect each parameter has then we can't use this information. Perhaps depth is much more significant than the other parameters so that they can be ignored on a first approximation?
Author Response
The authors would like to thank the reviewer for providing such an insightful review of the paper which has helped greatly in improving the paper. We have tried to answer the comments from the reviewer in detail.
-----------------------------------------------------------
This is a good paper.
What are the question marks for on line 191? Please fix
Author's Comment : We have fixed this typo.
One major concern is the water column variations, defined as temperature, salinity and depth.
It seems that depth was the only factor studied. Is this the case? If not, then data is needed on the temperature and the salinity. If so, then a section saying why they weren't examined or how the depth parameter was isolated is required.
Author's Comment : We agree with the reviewer that the water column variations are defined by temperature, salinity, and depth. But there are different natural phenomena in the water like internal waves, solitons, etc. These natural phenomena are studied in the literature using different techniques like SAR images or studying the data from dense thermistor chains in the water column. Firstly, in our experiment, we were not able to collect the salinity data. Secondly, we only have receivers at 30 m and 60 m depth which is not dense enough to study the water column variations. For this reason, we have done the Doppler analysis of the acoustic data and showed that it is possible to observe these variations using the signal processing tools. Another important factor is that these water column variations travel in the water column, and using only temperature or salinity data to detect them will only be effective once they pass the sensor. Using the acoustic tools we are able to detect these variations when they are present between the source and receiver because they affect the acoustic signal. We have not tried to isolate any single parameter and showed its effect on the acoustic signals. We have shown that acoustic tools can be used to study the presence of water column variations.
Your conclusion supports the theory that all three parameters affect the communications but without knowing what effect each parameter has then we can't use this information. Perhaps depth is much more significant than the other parameters so that they can be ignored on a first approximation?
Author's Comment : We have not tried to isolate the effect of three parameters on the communications. We were not able to collect the salinity data and the depth and temperature data were not dense enough to justify the presence of the water column variations. The major contribution of the paper is that the acoustic tools can help in detecting the water column variations and these water column variations have a great impact on underwater acoustic communications.
Reviewer 2 Report
In this paper, the effect of water column variations on the sound propagation and acoustic communications is studied. First of all, the channel IR estimates are studied and some of the environmental effects on the channel IR estimates are explained. Secondly, the time variability is studied in the channel IR estimates by computing channel coherence. Thirdly, the performance of the underwater communication system is shown in the paper.
The author has studied the effect of water column change on underwater acoustic communication, which has certain reference value. However, the overall innovation of the article is insufficient, only the analysis of the sea test data, and the conclusion of the data analysis is insufficient. .
Author Response
The authors would like to thank the reviewer for providing such an insightful review of the paper which has helped greatly in improving the paper. We have tried to answer the comments from the reviewer in detail.
--------------------------------------------------------------
In this paper, the effect of water column variations on sound propagation and acoustic communications is studied. First of all, the channel IR estimates are studied and some of the environmental effects on the channel IR estimates are explained. Secondly, the time variability is studied in the channel IR estimates by computing channel coherence. Thirdly, the performance of the underwater communication system is shown in the paper.
The author has studied the effect of water column change on underwater acoustic communication, which has certain reference value. However, the overall innovation of the article is insufficient, only the analysis of the sea test data and the conclusion of the data analysis is insufficient.
Author's Comment : In this paper, we have shown that acoustic tools can be used to study the water column variations. The water column variations include internal waves, solitons, etc which are studied in the literature using SAR images and temperature data. To our knowledge, acoustic tools are not used to study these effects. This study is very helpful in understanding these underwater phenomena. In addition to that, it shows that these water column variations have a significant effect on underwater communications.
Reviewer 3 Report
Please correct typos:
Line 97: "chrip" should be "chirp"
Line 98: "Binary phase shift keying" should be "Binary Phase Shift Keying"
Line 99: "chirp signal were transmitted" should be "chirp signals were transmitted"
Line 191: fix "Eq('??)"
Figures:
Figure3: what's the meaning of "absolute amplitude"? Amplitude in volts? in pascal?
Figures 4-6: merge each pair of subplots into one subplot with different markers
Figure 8: same as previous
Figure 9: confusing caption
Recommendation:
Add a ray trace (perhaps with PlaneRay?)
Author Response
The authors would like to thank the reviewer for providing such an insightful review of the paper which has helped greatly in improving the paper. We have tried to answer the comments from the reviewer in detail.
---------------------------------------------
Please correct typos:
Line 97: "chrip" should be "chirp"
Line 98: "Binary phase shift keying" should be "Binary Phase Shift Keying"
Line 99: "chirp signal were transmitted" should be "chirp signals were transmitted"
Line 191: fix "Eq('??)"
Author's Comment : All the typos are corrected.
Figures:
Figure3: what's the meaning of "absolute amplitude"? Amplitude in volts? in pascal?
Author's Comment : The Channel IR is computed by correlating the transmitted chirp signal with the received chirp signal. The absolute amplitude of the channel IR represents the value of the correlation between the two. We have changed the y-axis label to "Amplitude".
Figures 4-6: merge each pair of subplots into one subplot with different markers
Figure 8: same as previous
Author's Comment : Figs 4 and 5 are merged to one plot using different markers. The Doppler variations in Figs. 6 and 8 are plotted with the error plots. The error plots are chosen because the Doppler varies between a range of values. Error plot does not support different markers for the bars so it is not possible to merge all the plots together with different markers.
Figure 9: confusing caption
Author's Comment : The caption is edited as follows : "Bathymetry diagram showing the direction of the water column variation along the range. The white line represents the water column variation traveling in the water and the white arrows show its traveling direction."
Recommendation:
Add a ray trace (perhaps with PlaneRay?)
Author's Comment : We are not sure about the recommendation of the reviewer about adding the ray trace. It is not specified which scenarios should be modeled with the ray tracing.
Round 2
Reviewer 1 Report
I thank the authors for there response. I am now happy with the paper.
This manuscript is a resubmission of an earlier submission. The following is a list of the peer review reports and author responses from that submission.